



# The investigation of June 21 and 25, 2015 CMEs using EUHFORIA

Somaiyh Sabri[1] and Stefaan Poedts[2,3]

[1]Institute of Geophysics, Faculty of Physics, University of Tehran, Tehran, Iran
[2]Center for mathematical Plasma Astrophysics, Department of Mathematics, KU Leuven, Celestijnenlaan 200B, 3001 Leuven, Belgium
[3]Institute of Physics, University of Maria Curie-Skłodowska, ul. Radziszewskiego 10, PL-20-031 Lublin, Poland

**Correspondence:** Somaiyh Sabri (s.sabri@ut.ac.ir)

**Abstract.** In this research, the EUropean Heliosphere FORecasting Information Asset (EUHFORIA) is used as a mathematical model to examine how coronal mass ejections (CMEs) move through a solar wind flow that is not consistent in all areas, taking into account three dimensions and changes over time. Magnetohydrodynamic (MHD) simulations were conducted to analyze the propagation patterns of two specific CMEs that occurred on June 21 and 25, 2015. The EUHFORIA simulations
for the inner region of the heliosphere involve incorporating conditions related to CMEs and the solar wind at the boundaries. Comparative examination using data from the WIND and OMNI spacecrafts reveals that the EUHFORIA model offers a moderately precise depiction. The study highlights that interactions of CMEs play a significant role in determining their impact on Earth, highlighting that their initial speeds, while similar, are less influential. Besides, the EUHFORIA numerical model align with the findings of the GFZ German research center, this implies that EUHFORIA has also the capability to compute
and potentially forecast the impact of CMEs on the Earth.

## 1 Introduction

Space weather refers to the impact of solar activity on Earth and other celestial bodies in the solar system. The scientific community recognizes the increasing significance of studying space weather due to its impact on human activities. Actually, various environments require distinct principles of physics to be utilized. To address this, the ESA Virtual Space Weather Modeling
Center (VSWMC) has expanded its capabilities to allow the integration of a series of models for the purpose of forecasting Poedts et al. (2020). The current model for the development of heliospheric wind and CMEs, such as the coronal model in EUHFORIA, relies on the Wang-Sheeley-Arge (WSA) model Arge et al. (2003). It is important to highlight that alternative coronal model and Multi-VP need more CPU time Reville et al. (2015); Samara et al. (2021).

The release of a significant amount of magnetic energy can occur when there are rapid changes in the magnetic structures through a process called magnetic reconnection McLaughlin et al. (2018); Sabri et al. (2018, 2019, 2020a, b, 2021a, b, 2022, 2023); Kumar et al. (2024). CMEs occur when a large quantity of plasma from the sun's corona is expelled into space. They are explosive events on the Sun where plasma and magnetic fields are forcefully ejected and form significant structures. These structures,



along with their shocks, travel through the heliosphere. They play a crucial role in linking solar eruptions to the subsequent
effects on interplanetary and geomagnetic disturbances Dryer (1994).

In general, the majority of fast reverse shocks are generated by interactions between slow and fast stream regions, while
fast forward shocks are primarily caused by CMEs throughout all stages of solar activity except during periods of solar mini-
mum Kilpua et al. (2015). Fast collisionless shocks are significant solar occurrences. The steepening of fast magnetoacoustic
waves is connected to the sudden increase in both the magnetic field strength and the plasma parameters of the solar wind.
This steepening process may be responsible for the creation of fast shocks. Studying the significance of fast shocks in solar
terrestrial physics is crucial as it leads to the acceleration of charged particles to high energies (around tens of Mev). They pose
significant risks to satellite technology and human ventures in space Manchester et al. (2005).

Since major geomagnetic storms are driven from CMEs, to forecast their arrival at Earth or any planets or satellites, it is
significant to investigate their start and also propagation. Space weather forecasting is dependent on different factors includ-
ing the eruptions at the Sun, their moving from the solar corona to the planets or satellites in the inner heliosphere Riley &
Ben-Nun (2021); Verbeke et al. (2022). In this line, magnetohydrodynamic simulations are helpful to investigate the prop-
agation of CMEs, study their interactions with solar wind structures and also other CMEs and computing their geoeffectiveness.

Understanding three-dimensional background solar wind through numerical analysis is crucial for accurately predicting and
studying space weather phenomena, specifically for identifying the specific behavior of three-dimensional CME propagation.
Successful combining of two dimensional and three-dimensional MHD coronal and heliospheric models has been performed
by Wu et al. (1999); Odstrcil et al. (2004). The coronal simulation began with an initial potential magnetic field and a spher-
ically symmetric Parker solar wind Odstrcil et al. (2002). The coronal model utilized a series of MHD flow parameters over
time, which served as a boundary condition for the heliospheric model.

The objective of our study is to investigate the propagation and interaction of specifically identified CMEs with other CMEs
as well as the solar wind. In this research, we selected two CMEs and examined their propagation using 3D MHD modeling
EHUFORIA. Our goal was to determine the timing of their arrival at Earth and assess whether they would cause geomagnetic
storms or not.

## 2  Physics of coronal mass ejections

The occurrence of CMEs is typically connected to the disruption of a coronal magnetic structure that resembles a partially
toroidal magnetic coil, which is anchored in two opposite polarities of the sun's magnetism on its surface Torok & Kliem
(2005); Schmieder et al. (2015); Green et al. (2018). This explanation pertains to the requirement of the presence of magnetic
energy and electric currents in the coronal structure for its acceleration into the corona and interplanetary space. In relation
to the magnetic field configuration prior to an eruption, it is important to note that the cause of the eruption cannot solely be





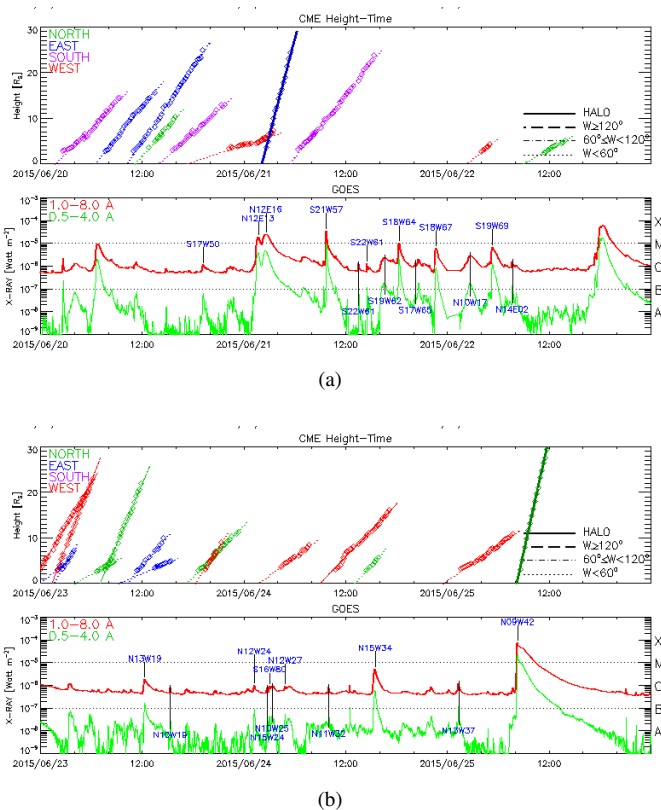

**Figure 1.** Overview of the early evolution and eruption of CME1 and CME2. Panels (a) and (b) span the time ranges of June 20-25, 2015. Panel (a) illustrates the Height-time plot of CME1 and also M-class flare on June 21, 2015. Panel (b) depicts the Height-time plot of CME2 and also M-class flare on June 25, 2015. http://ipshocks.fi/database

attributed to a flux rope. Other factors such as a sheared arcade or multiple quadrupolar structures could also lead to eruptions Song et al. (2014); Nindos et al. (2020). It must be noted that the eruption is pioneered by swelling and slow increasing motion of the pre-eruptive structure.


The reduced solar activity in cycle 24 leads to an increase in the speed of coronal mass ejections. In this scenario, the majority of the CMEs manifest as halo CMEs (HCMEs), which are formed closer to the sun. The Solwind coronagraph on the P78-1 spacecraft was the first to identify this type of CMEs Howard et al. (1985). Information gathered from the coronagraphs on the Solar and Heliospheric Observatory (SOHO) and the Solar Terrestrial Relations Observatory (STEREO) has revealed that

halo CMEs are a type of CME that are generally fast and wide Gopalswamy et al. (2013); Makela et al. (2016). The crucial matter is that the halos contain a significant number of highly energetic CME populations. Almost 58% of CMEs result in low-frequency type 2 radio bursts while 67% of CMEs cause intense geomagnetic storms (DSt$< -100nT$) and around 80% in CMEs related on the large solar energetic particle (SEP) events and 100% associated with gamma-ray emission (Gopalswamy





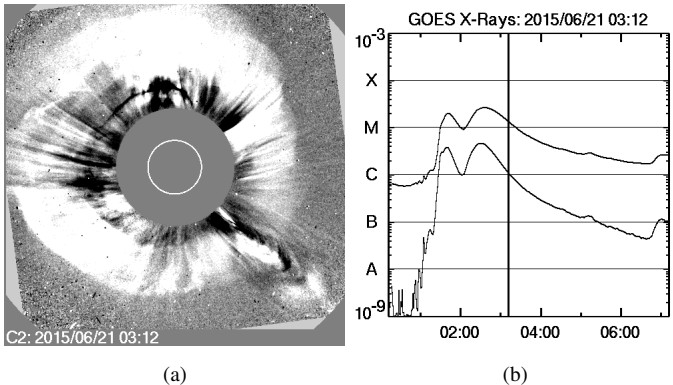

(a)        (b)

**Figure 2.** CME1 and flare development on June 21, 2015, in the LASCO C2 field of view and GOES X-ray respectively. This figure is based on the SOHO LASCO CME CATALOG, which can be accessed through the following link. https://cdaw.gsfc.nasa.gov/CME_list/

et al. , 2018, 2019b).


In order to have a halo shape, CMEs must begin near the central meridian of the Sun, either in the front or in the back as distinguished from the coronagraph. Roughly $10\%$ of halo CMEs originate from the limb and these are particularly fast, with a speed of $(1400kms^{-1})$ (Gopalswamy et al. , 2020; Cid et al. , 2012). There is a significant probability that a frontal halo CME observed from the line connecting the Sun and Earth will collide with the Earth and result in a geomagnetic storm Zhao & webb (2003); Scolini et al. (2018). During solar cycle 24, the number of halo CMEs remained relatively stable, despite a decline of over $40\%$ in the sunspot number (SSN). This observation is supported by studies conducted by (Michalek et al. , 2019; Dagnew et al. , 2022). Another aspect to consider is that the reduced pressure in the heliosphere during Solar Cycle 24, as compared to Solar Cycle 23, led to the expansion of CMEs. These CMEs could be classified as halos, even if they originated from areas further away from the central meridian distance (CMD).


## 3 Observation of the 17-27 June 2015 CMEs

The analysis of a CME involves assessing its size, speed, and direction, which are critical parameters in understanding its behavior. SWPC forecasters use coronagraph imagery from orbital satellites to determine the probability of any potential impact on Earth based on specific characteristics. The Large Angle and Spectrometric Coronagraph (LASCO) is a coronagraph installed on the NASA Solar and Heliospheric Observatory (SOHO). This device has two capabilities for capturing images of the Sun's corona using optics. One range, called C2, can capture images within a distance range of 1.5 to 6 times the radius of the Sun, while the other range, called C3, can capture images within a distance range of 3 to 32 times the radius of the Sun. The LASCO instrument is presently the main tool utilized by forecasters to examine and classify CMEs. Nevertheless, there is



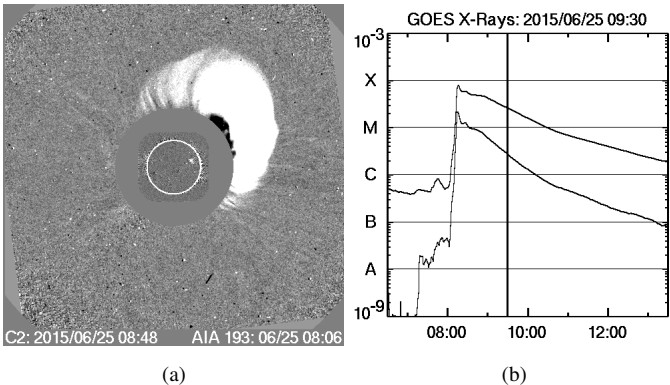

(a)          (b)

**Figure 3.** CME2 and flare generation on June 25, 2015, in the LASCO C2 field of view and GOES X-ray respectively. This figure is based on the SOHO LASCO CME CATALOG, which can be accessed through the following link. https://cdaw.gsfc.nasa.gov/CME_list/

another coronagraph on the NASA STEREO-A spacecraft that serves as an extra resource.


In this portion, we aim to examine the observable signs of two halo CMEs that occurred on June 21 and 25, 2015. Figs. 1 shows the occurrence of two halo coronal mass ejections. The initial CME occurred on 21 June 2015 at around 03:00 UT with velocity $1366km/s$ was associated with M2.6 flare situated at N12w08. CME2 commenced on June 25, 2015 on 08:36 UT with a velocity of $1627km/s$ that linked to an M 7.9 flare, which was situated at N09w42. In each panel, first image demonstrates

the kind of the flares and X-ray intensities related on two CMEs eruptions. These eruptions were detected in the field of view of the C2 instrument of Large Angle and Spectrometric Coronagraph (LASCO) on the Solar and Heliospheric Observatory (SOHO) and also COR-2 instrument on board the Sun-Earth Connection Coronal and Heliospheric Investigation (SECCHI) on the twin-spacecraft Solar Terrestrial Relations Observatory (STEREO) Brueckner et al. (1995); Kaiser et al. (2008).

In Figs. 2, CME1 was detected in the C2 FOV at 03:12 UT with the following M2.6 flare that is in agreement with the results in panel (a) of Fig. 1. Beside, panel (a) of Fig. 2 depicts that this CME could be considered as full halo CME. These types of CMEs usually take place at the area of the Sun with localized and strong magnetic field including active regions. Moreover, in Fig. 3 CME2 was also viewed in the C2 FOV at 08:06 UT with its following M 7.9 flare in according to the panel (b) of Fig. 1. The shape of the CME2 that is observable in panel (a) of Fig. 3 illustrates a kind of limb CME.


In according to panel (a) of Fig. 4, CME1 that happened on 21 June, 2015 has the linear fit speed 1366 $km/s$. As we know, CMEs move outward from the Sun at speeds ranging from slower than $250km/s$ to as fast as almost $3000km/s$. The fastest Earth-directed CMEs can arrive our planet in around $15-18hours$. Then, this CME with speed $1366km/s$ is a kind of fast CME and should arrive at Earth near one day. In the case of the second selected CME or CME2, it has the linear fit speed

$1626km/s$ then it is somehow faster than the first CME. It could be expected that CME2 may result in the significant storms



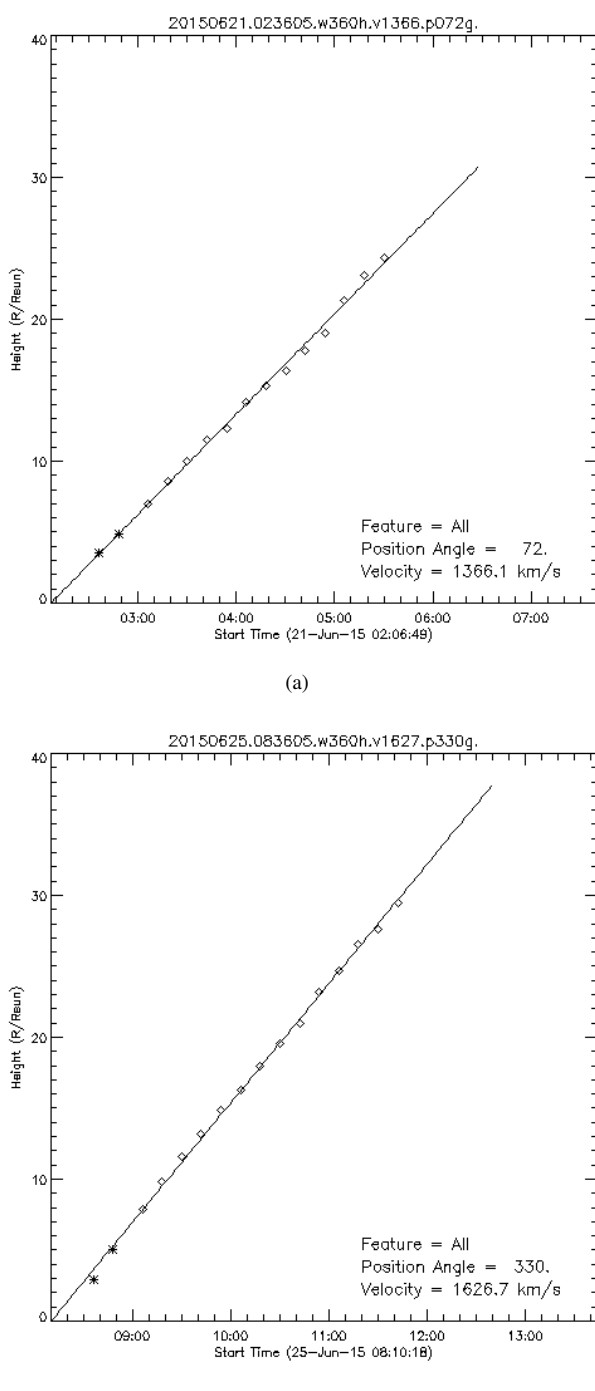

**Figure 4.** CME speeds with linear fit and also second order fit for both two CMEs that happened on 21 June, 2015 and 25 June,2015. Panel (a) shows the linear fit velocity and panel (b) demonstrates the second order fit for the first CME happed on 21 June, 2015 and Panel (c) shows the linear fit and panel (d) defines the second order fit velocity for the second CME happed on 25 June,2015. This figure is based on the SOHO LASCO CME CATALOG, which can be accessed through the following link. https://cdaw.gsfc.nasa.gov/CME_list/



on the Earth.

CMEs that are more intense usually occur when magnetic field structures or flux ropes in the lower corona of the Sun, which are highly twisted, become disrupted and change their shape to a less tense configuration. This phenomenon is referred to as magnetic reconnection. This occurrence can be accompanied by the sudden release of electromagnetic energy in the form of a solar flare. As demonstrated in panel (b) of Figs. 2 and 3 these CMEs were accompanied by M-class solar flares.

Fig. 5 illustrate the distinct characteristics of the Coronal Mass Ejections. The shocks caused by CMEs are represented by the red line in every panel. The shock is identified by the increase in both velocity and number density. The shock candidates were detected using a visual inspection technique. In this approach, we examined daily graphs of solar wind plasma and magnetic field characteristics in order to identify potential instances of shock. In fact, simultaneous and abrupt changes in both plasma and magnetic field factors were taken into account. These jumps had to be considerable enough and fulfill the properties of either fast forward (FF) or fast reverse shock (FR).

Fast shocks can be categorized into two types: fast forward shocks, which move in the opposite direction away from the Sun, and fast reverse shocks, which move towards the Sun while exhibiting a significant outflow of solar wind from the Sun. In the solar wind, when fast forward shocks pass over the spacecraft, there is a simultaneous rise in the magnetic field and all plasma parameters. This is depicted in Figs. 5. On the other hand, in the case of fast reverse shocks, the speed increases but the density of the solar wind, magnetic field strength, and temperature all decrease. In comparison to CME1, CME2, which is depicted in Figs. 5, has a substantial increase in both velocity and magnetic field strength. This could potentially result in a severe storm. Furthermore, the notable values of $N_p$ and $T_p$ on the second panel for CME2 in contrast to CME1 also confirm the possibility that CME2 could result in a severe storm. This will be elaborated further in the subsequent discussion.

## 4 Numerical results and discussion

### 4.1 EUHFORIA

EUHFORIA is divided into two crucial sections. One approach is the Wang-Sheeley-Arge model (WSA), which incorporates data from magnetograms to determine the plasma conditions at a distance of 0.1 AU. These conditions are necessary for predicting the background solar wind in the heliosphere McGregor et al. (2011); van der Holst et al. (2010). Various magnetograms can be employed, such as the Global Oscillation Network Group (GONG; Harvey et al. (1996)) or the Helioseismic and Magnetic Imager (HMI) of the Solar Dynamics Observatory (SDO; Schou et al. (2012)). To begin with, in this model of the Sun's outer atmosphere, the three-dimensional magnetic field in the corona is determined by using a method called Potential Field Source Surface (PESS) extrapolation, which was introduced by Altschuler & Newkirk (1969). Next, the magnetic field is stretched to a distance of 0.1 astronomical units (AU) utilizing the Schatten Current Sheet model as mentioned in (Schatten





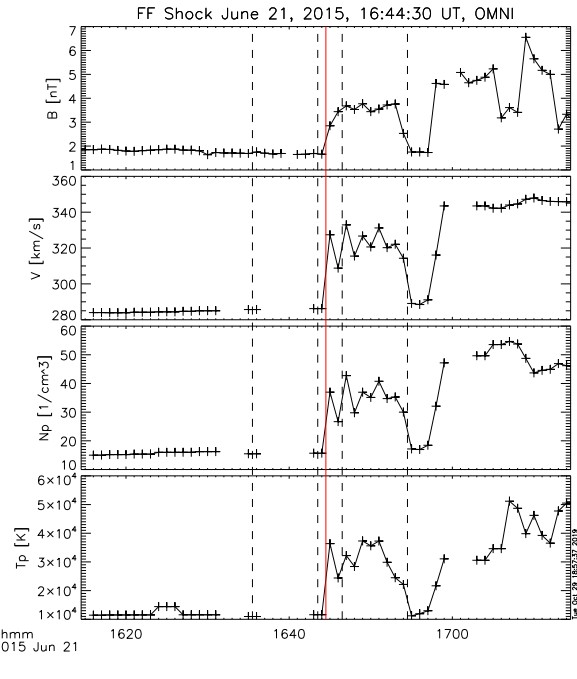

(a)

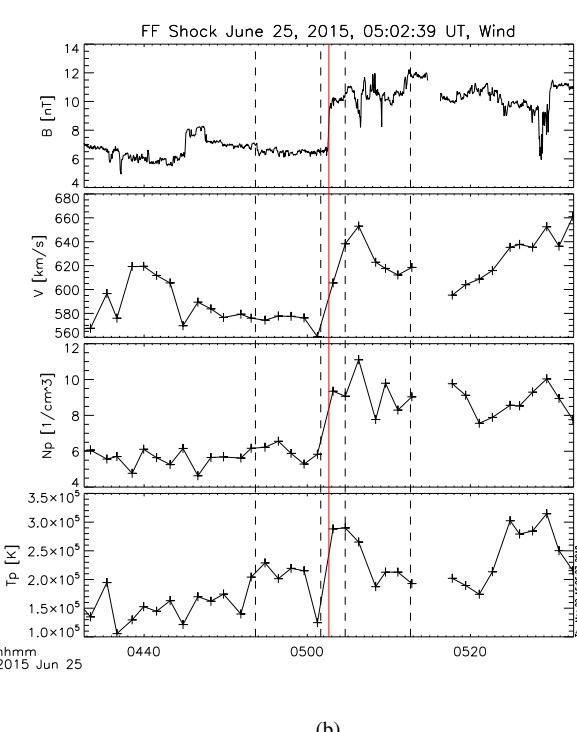

(b)

**Figure 5.** Fast forward (FF) shocks are demonstrated by OMNI and Wind spacecrafts on 21 and 25 June 2015 that include total magnetic field $B$, the magnitude of the bulk velocity $V$, the proton density $N_P$ and the proton temperature $T_P$. http://ipshocks.fi/database



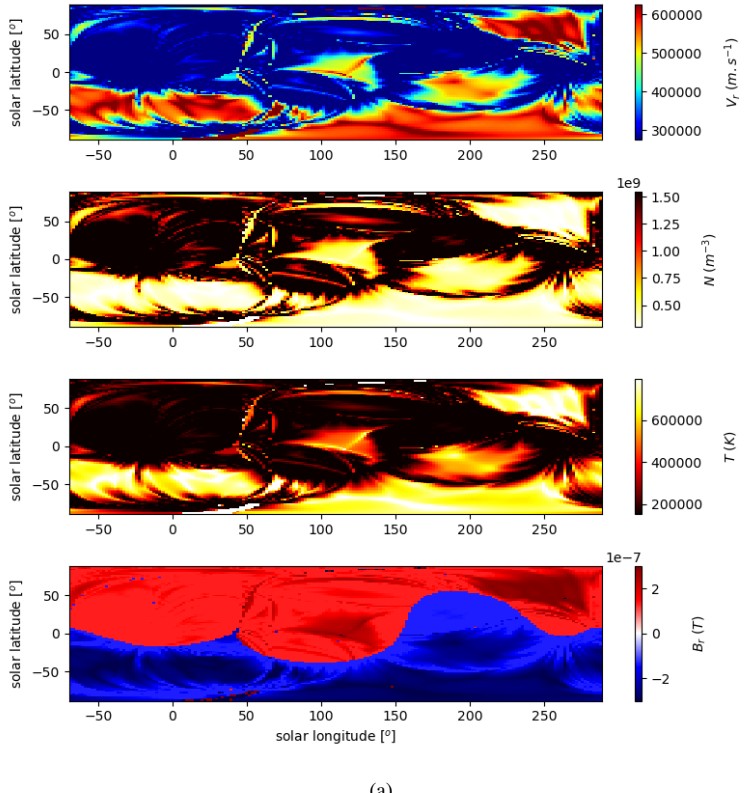

(a)

**Figure 6.** The EUHFORIA input boundary condition that shows distribution of the radial velocity $(m/s)$, number density $(1/m^3)$, temperature $(K)$ and radial magnetic field $(T)$.

et al. , 1969). At the 0.1 AU limit, scientists determine the speed of the solar wind. Using this speed, they calculate the density

and radial velocity at the outer boundary by employing empirical relationships Pomoell & Poedts (2018).

The next stage in EUHFORIA involves conducting a simulation that models the heliosphere, taking into account its 3D nature and also time-dependent. It is important to note that the thermodynamic and magnetic measurements were calculated using the coronal model during an MHD relaxation phase with a uniform grid, specifically within the range of 0.1 AU and 2

AU. This sentence states that a finite volume numerical method and a constrained transport scheme are used to solve the ideal MHD equations with a polytropic index of 1.5, ensuring that the solenoidal criteria are met.

In this research, we examined the development of eight specific CMEs that occurred between June 17 and June 29, 2015. Certain CMEs have distinct features: they are rapid, with speeds exceeding $500km/s$, and they are not narrow, meaning their

half-widths are greater than 35 degrees. The most crucial characteristic is that these selected CMEs are aimed at the Earth, with their source longitudes falling within a range of approximately plus or minus 60 degrees HEEQ. The solar wind was introduced



by the Global Oscillation Network Group (GONG,Harvey et al. (1996)) at 00:03 UT on June 17, 2015. This information was randomly chosen to showcase the potential application of EUHFORIA.

GONG defines map products with 360 pixels in the east-west direction and 180 pixels in the north-south direction. There are a total of five different synoptic maps found in the GONG products. Two vital magnetogram synoptic maps are being created by utilizing data from the entire Carrington rotation. The complete magnetic field of the Sun is determined by utilizing accurately calibrated one-minute full-disk magnetograms from GONG's six regions.

Figs. 6 displays the magnetic map utilized as an input for the coronal portion of EUHFORIA. Based on the significant disturbances in parameters, we can infer that the Sun is approaching a period of maximum activity. Fig 6 displays the most recent prediction of the radial velocity, density, temperature, and radial magnetic field of solar wind.

      The primary objective of this study was to assess the similarities and differences between observational data and the EUHFO-
RIA simulation, a new space weather forecasting model for the inner heliosphere. EUHFORIA focuses on the characteristics of solar wind near the Sun, as well as the transient events related to CMEs expanding into the heliosphere. The inputs of the model include a corresponding file containing information about the magnetogram for the Corona model, as well as CMEs detected through observations from a coronagraph.

The part of EUHFORIA known as the heliosphere module focuses on studying the solar wind that spans from 21.5 Rs to 2 AU Pomoell & Poedts (2018). A coronal module applies an input of 21.5Rs. The text describes the process of expanding the radial velocity and magnetic field of a system to a distance of 2 AU. To ensure that the solar background wind is in a relaxed state, the initial MHD solution is further altered by including a rotating inner boundary. As a consequence, there is a continuous flow of solar wind from 21.5 Rs to 2 AU in the co-rotating frame because the inner boundary condition remains unchanged
and is only rotated. The equations are solved in the HEEQ frame, which is defined as the frame with its Z-axis directed in the rotation axis of the Sun, and its X-axis considered by the intersection of the solar equatorial plane and the solar central meridian of date that seen from the Earth.

      Furthermore, when simulating the interaction between CMEs and solar wind topologies, it is often necessary to take into
account not only the surrounding ambient background but also the ejecta itself, which typically includes an internal magnetic field. This is done to ensure more accurate and realistic simulations, as opposed to using a simpler hydrodynamic pulse. However, simulations involving a magnetized ejecta can be computationally demanding when applied to Sun-to-Earth simulations Jin et al. (2017); Torok et al. (2018). The velocity and density of the CMEs are associated with the dynamic pressure and also include sign of the moving of the interplanetary shock at the front of the CMEs. Because geomagnetic events can be caused
by interplanetary shocks alone, the simplest models do not consider the internal magnetic field topology of the coronal mass





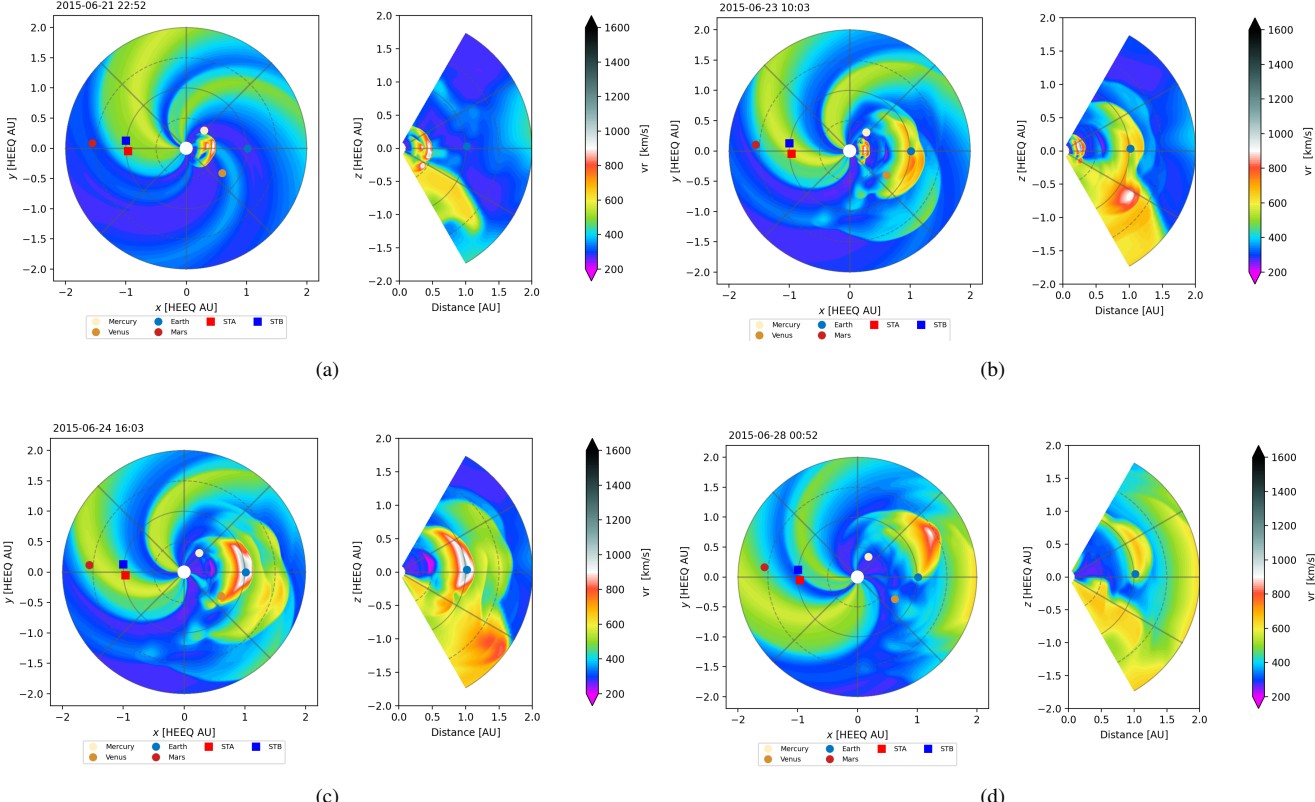

**Figure 7.** Snapshot of solar wind radial velocity ($km/s$) from the MHD simulation with EUHFORIA. In each row, the left images depict the solution in the heliographic equatorial plane and the right panels demonstrate the meridional plane that includes the Earth.

ejection (CME)Oliveria & Samsonov (2018).

The EUHFORIA-heliosphere model can also simulate the launch of CMEs while taking into account the background solar wind. As a result, it can simulate CME evolution up to 2 AU and beyond. Most of the CMEs expand outward in a radial

manner beyond the first few solar radii Plunkett et al. (1997). Because of the specific qualities observed in CMEs, the initial assumption about the cone-shaped halo-CMEs was modified by including additional parameters. These parameters determine the angle of the cone and the direction in which its central axis is oriented Zhao et al. (2002). They utilized uncharged variables to determine the angular breadth of the cone and the alignment of the central axis of the cone. It is important to mention that the cone model was also utilized in the numerical simulation conducted byOdstrcil et al. (2004). The classic cone CME model

is completely validated in the European Heliospheric Forecasting Information Asset (EUHFORIA) system.



Furthermore, taking into account the movement of CMEs, particularly in complex situations where there is interaction with the solar wind, poses a significant challenge for empirical approaches. Magnetohydrodynamic simulations provide a practical method to utilize physics-based modeling. The ENLIL model is the sole MHD lithospheric method utilized for forecasting, according to Parsons et al. (2011).

Despite the community's extensive effort to predict arrival time of the CMEs, their accuracy remains uncertain. Both empirical and physics-based approaches have indicated that the root-mean-square error in predicting the arrival is 12h and 10h respectively Zhao & Dryer (2014); Mays et al. (2015). The primary concern is that different CMEs have varying levels of impact on Earth. Therefore, it is crucial to enhance our capability to predict how these events will impact Earth. It is important to highlight that the magnetic field of CMEs is crucial in determining their impact on Earth and our ability to understand the characteristics of the magnetic field is very limited. Only methods related to space weather are used to predict the magnetic structure of CMEs Isavnin, (2016); Kay et al. (2017).

In this part, the outcomes of the EUHFORIA simulation for the inner heliosphere from 17 to 29 June, 2015 are illustrated. The application includes the implementation of boundary conditions involving CMEs and solar wind. The study aims to examine the propagation of both CMEs.

Fig. 7 shows the representation of the heliospheric component of EUHFORIA, specifically focusing on the simulation of the underlying solar wind. The speed chosen is radial and matches the empirical speed recommendation. As depicted in Fig. 7, the size of the CME gradually increases as it travels. While the CME is spreading, the thermal pressure causes it to consistently expand. The magnetic forces in the axial direction slow down the motion in the radial direction. CME magnetic pressure changes depending on the solar wind, causing the magnetic pressure to alternate between shrinking and expanding. The initial purely radial speed of the CME center continues to propagate in the subsequent time step. The expansion of the CME is caused by two main forces, which are the Lorentz force and the drag force. The Lorentz force, denoted as $J \times B$, is equivalent to the multiplication of the current density ($J$) and the magnetic field ($B$). This force allows for the radial expansion of the CME Sachdeva et al. (2015). However, the drag force has a tendency to impede the movement of the CME and is responsible for its gradual deceleration as it propagates Chen & Kunkel (2010); Subramanian et al. (2012); Sachdeva et al. (2017). The acceleration of the CME occurs when the Lorentz force becomes dominant over the drag force due to the development of the magnetic field.

To effectively predict space weather, it is crucial to model the movement of CMEs in the interplanetary space, taking into account the background solar wind. This approach ensures accurate determination of the ejection's properties and enables the estimation of when the CMEs will reach Earth MacNeice et al. (2018); Reiss et al. (2022). It should be noted that when simulating the arrival time, errors of approximately 1 day are commonly observed Jian et al. (2011); Gressl et al. (2014). Despite this, many studies have focused on assessing how different models, including empirical/semi-empirical, machine learning, and MHD approaches, can account for the solar wind conditions in the surrounding environment Barnard & Owens (2022);



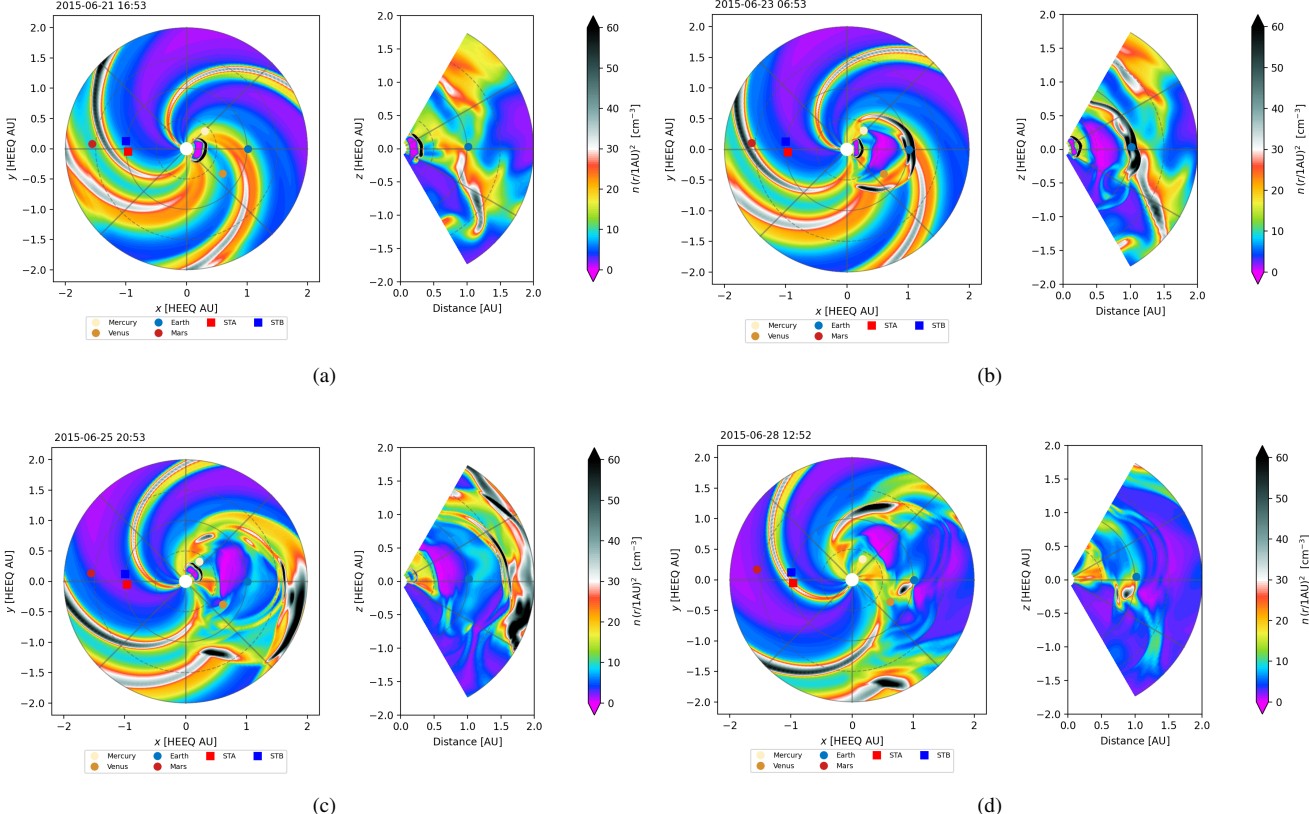

**Figure 8.** Snapshot of solar wind particle density ($1/cm^{-3}$) from the MHD simulation with EUHFORIA. In each row, the left images depict the solution in the heliographic equatorial plane and the right panels demonstrate the meridional plane that includes the Earth.

Milosic et al. (2023); Huang et al. (2023).

We determine that CME1 will reach the Earth in approximately 46 hours based on the speeds projected in the CDAW cat-
alogue (https://cdaw.gsfc.nasa.gov/CME-list/). CME1 has a linear fit speed of 1366.1 km/s, which means it will take around
30.49 hours for it to reach the Earth. Actually, the second estimate is a rough approximation that does not take into account
the impact of drag or the interaction between CMEs as they travel towards Earth. These factors could explain the discrepancies
observed. In the situation of CME2, this distinction becomes notable. The estimated arrival time of CME2 using EUHFORIA
results is approximately 70 hours, whereas using the linear fit speed of 1626.7 km/s, the estimated arrival time is around 26
hours. It can be inferred that the way in which CMEs are launched is not particularly significant, but their interaction and move-
ment in the heliosphere are the main factors in determining their impact on Earth. In fact, Figs. 7 and 8 illustrate the growth of
CME1 while it travels, while CME2 has diminished as it traveled, suggesting that CME2 has lost its energy. Afterwards, it was



anticipated that a less powerful CME2 would reach its destination later than a stronger CME1, aligning with the calculated data.

Fig. 8 presents four snapshots of plasma density distribution. The plasma number density is defined by the following equation.

$$n_{scaled} = n \, \frac{r}{1AU}^2 . \tag{1}$$

In each plot, the left panel shows the amount in the heliographic equatorial plane, whereas the following panel displays the density in a meridional plane that includes Earth. In addition, the circles are shown with heliocentric radii set at values
of $r = 0.5, 1, 1.5, 2AU$. It is important to mention that the positions of the inner planets and the locations of both STEREO spacecrafts are indicated using markers. The information depicted in Fig. 8 indicates that the near-Earth environment experiences substantial changes in its dynamics caused by frequent eruptions. The panel (b) illustrates the arrival of CME1 at Earth, revealing that the majority of the CME reaches our planet. Panel (d) of Fig. 8 depicts the arrival of CME2 at the Earth's profile, demonstrating that only a section of CME2 reaches the Earth. Subsequently, it is anticipated that the first coronal mass ejection
(CME1) will have more substantial repercussions on Earth when compared to CME2, which will be further elaborated on.

The study conducted by Burlaga & Ogilvie (1969) revealed that changes in the density of materials within the geomagnetic field caused stress, rather than the occurrence of shocks. In addition, Rufenach et al. (1992) demonstrated that when solar wind dynamic pressure increases, there is typically a corresponding increase in the average magnetic field of the magnetosphere as
measured by the GOES satellites in geosynchronous orbit. Due to the significant fluctuations in plasma density, it can be inferred that CME1 exerts greater pressure on the Earth's magnetosphere compared to CME2, which exhibits fewer plasma density fluctuations upon arrival.

The standardized K index of 13 magnetic observatories was used to derive the geomagnetic Kp index Bartels & Veldkamp (1950). This particular measurement assesses solar particle radiation by examining its magnetic reactions and determines the
magnitude of geomagnetic storms. The K-index measures disruptions in the Earth's magnetic field in the horizontal direction, and it is assigned a value from 0 to 9. In calm situations, the Kp value is approximately 1, whereas during intense geomagnetic storms, it is roughly 9. The Kp index is a global measure of the K index, which is determined by analyzing data from magnetometers located on the ground Bartels & Veldkamp (1949). Indeed, when it comes to the reality of the situation, Kp serves as a remarkable gauge for abnormalities within the magnetic field of the Earth. Presently, Kp holds significance as it quantifies
the energy transfer from the solar wind to Earth and is employed by space weather services in almost real-time.

EUHFORIA utilizes a straightforward approach that is based on the empirical equation for linear prediction of Kp, which was originally proposed by Newell et al. (2008). They discovered that the predictability of Kp can be greatly enhanced by incorporating both a merging factor and a viscous factor when analyzing solar wind data. The solar wind information used for
predicting the Kp index can be obtained, for instance, by consulting the forecast outputs of EUHFORIA at Earth. The Kp index is computed by the model and presented as a time series file. Fig 9 shows the changes in the Kp index over time, as calculated




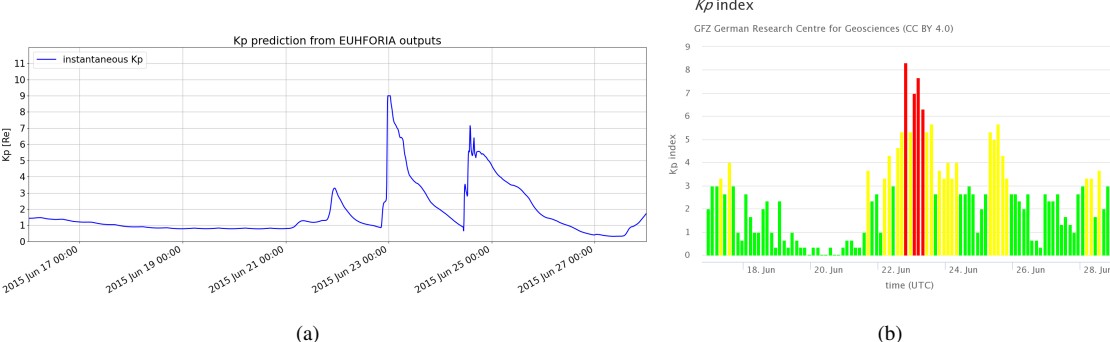

(a)          (b)

**Figure 9.** Kp index calculated by the EUHFORIA and observational GFZ German research center.

by EUHFORIA in panel (a) and defined by the GFZ research center in panel (b). The results from the EUHFORIA numerical model align with the findings of the GFZ German research center, demonstrating that the Kp index exhibits consistent changes over time. This implies that EUHFORIA has the capability to compute and potentially forecast the impact of CMEs on the Earth. The noteworthy aspect is that the previous findings demonstrating the significant role of CME1 in geoeffective outcomes align with the information presented in Fig 9. Due to the information presented in Fig 9, it can be observed that the $K_p$ level is approximately 9 during the arrival of CME1. This signifies that CME1 leads to a severe geomagnetic storm. On the other hand, CME2 has a $K_p$ value of around 3 during its arrival, indicating a moderate storm.

## 5 Conclusions

CMEs are responsible for causing significant geomagnetic storms and generating solar energetic particles (SEPs). Because of their significance in both science and society, studies on CMEs become highly important. Geomagnetic storms have the potential to generate particles in radiation belts, which can impact satellites, while SEPs have the capability to harm spacecraft. Consequently, it is crucial to examine the characteristics of CMEs, such as their movement and interaction while being influenced by the surrounding solar wind. Understanding their arrival time at Earth is imperative for accurately predicting space weather.

We utilized a numerical MHD model with three-dimensional time-dependence to study expansive solar wind formations and explore the movement of certain CMEs and their propagation.

First, we examined a series of observational records that portrayed the situation. CMEs occurred on both June 21 and June 25, 2015. Out of the various CMEs that happened during that specific time, only two were chosen. These two CMEs had distinct structures, one being a full halo CME and the other being a limb CME. It is important to mention that although they



had similar speeds, there were other coronal mass ejections (CMEs) occurring around CME1. These interactions between the
CMEs could have had a notable impact on Earth's magnetosphere and ionosphere.

Upon reviewing the observational data, we utilize the EUHFORIA simulation to differentiate the actions of both CME1 and
CME2. Using this simulation, we discovered the manner in which these CMEs move and the precise moment of their arrival
at Earth's magnetosphere. Additionally, the geomagnetic parameter $K_p$ was graphed and compared to the observed value in
order to speculate which CMEs could lead to significant storms on Earth.

In conclusion, we have achieved the following key findings.:

1. In our research, we discovered that the "EUHFORIA" Heliosphere module, a three-dimensional MHD model, shows
reasonably similar results to the WIND and OMNI observations with including self-consistent structures.

2. It has been uncovered that as CMEs move through space, they gradually increase in size as a result of thermal pressure,
the Lorentz force, and drag force.

3. Snapshots of the plasma density distribution illustrate that dominant part of the CME1 arrives at the Earth while, just
flank of CME2 arrives the Earth. Additionally, CME1 generates notable plasma density pressures upon arrival, in contrast
to CME2. Therefore, it is anticipated that the impact of CME1 on Earth will be much greater than that of CME2.

4. The EUHFORIA model as well as observational data are utilized to calculate the arrival time of both CMEs at Earth.
It was shown that there are significant discrepancies in the estimated arrival time of CME2 when calculated using
the two mentioned methods. It can be inferred that the initial traits of CMEs are not highly significant, but it is the
interaction between CMEs and their movement through the heliosphere that primarily influences their impact on Earth.
This statement suggests that CME1 became stronger as it propagated, while CME2 became weaker. Consequently, it can
be inferred that CME2 lost energy and reached Earth at a later time.

5. The EUHFORIA numerical model's time variations of the Kp index coincided with the observational GFZ results. This
states that EUHFORIA has the capability to accurately determine and possibly forecast the impact of CMEs on Earth.
Furthermore, due to the presence of several CMEs in the vicinity of CME1, it is possible that this anticipated severe
storm is connected to the interaction of multiple CMEs.

We inferred that the solar wind in the background, which was identical in both CMEs, did not have a significant impact on the
creation of magnetic storms. It can be inferred that the nature of CMEs and particularly their interaction with other CMEs is
the primary factor determining the geomagnetic impact of CMEs on Earth.





*Code and data availability.* Observatory data used in this study was obtained from http://ipshocks.fi/database, https://cdaw.gsfc.nasa.gov/CME-

list. This CME catalog is generated and maintained at the CDAW Data Center by NASA and The Catholic University of America in cooperation with the Naval Research Laboratory. SOHO is a project of international cooperation between ESA and NASA. Besides, we used EUHFORIA code to numerical study that is publicly available vie the VSWMC in https://euhforia.com/.

*Author contributions.* Authors have contribution to write the manuscript.

*Competing interests.* The contributer author has declared that none of the authors has any competing interest.

*Acknowledgements.* For the computations we used the infrastructure of the VSC−Flemish Supercomputer Center, funded by the Hercules foundation and the Flemish Government−department EWI.



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
