# Peer review of "The investigation of June 21 and 25, 2015 CMEs using EUHFORIA"

_EGUsphere, 2024_

## Author Comment (AC1)

The article titled – 'The investigation of June 21 and 35, 2015 CMEs using EUHFORIA' models 2 CMEs that impact Earth and calculate the Kp index of the associated storm comparing it with the GFZ predictions'. The work has some merit in Kp prediction; however, the article lacks clear details of the numerical investigation and does not provide scientific evaluation of the CME evolution. The article is also not well written and touches on ideas without clear explanations or examples/citations. A major rewrite is needed. The comments below have addressed these shortcomings.

Specific comments:

Section 3 is titled 'June 17-25 CMEs' without any mention of the other CMEs associated with the CMEs studied in this work. According to the SOHO LASCO CME Catalog, there are multiple halo CMEs during June 17-25 including fast CMEs on June 18, June 19 before the studied June 21 and 25 events. On Line 155, the authors state that they examine the 8 specific CMEs, without any mention of these CMEs in the article. In fact, the authors only describe 2 CMEs on June 21 and June 25. Nowhere these 8 CMEs are described, identified, or even modeled. Were these CMEs geoeffective? Did they interact with the CMEs 1 and 2 causing a strong storm? The authors state that they examined the developed of these CMEs that impacted Earth, but it is not described in the text. The results of modeling 2 CMEs match with the Kp index, but the model has used only 2 events out of the mentioned 8 – does it mean that the interaction of the 2 CMEs has been overpredicted? Do the remaining 6 have no impact at earth? Why were they excluded?

**Our reply:** I apologise for my bad writing, which resulted in a misunderstanding. We considered the CMEs that happened on 18 and 19 June. I added Table 1 with the CMEs information in the manuscript. Because we had those CMEs on 18 and 19 June that happened before the CME1 (which occurred on 21 June), but we did not have significant CMEs on 23 and 24 June before our second selected CME2, we considered that for CME1, we had the CME interaction, but this did not happen for the CME2. Then, we investigated the CME interaction effect for CME1 that affects its impact on Earth. There are no CME interactions for CME2.

Line 93 – The CME 1 is shown to be associated with M2.6 flare at N12W08 – which is compared to Figure 2 and said to agree. However, nowhere on Figure 1 Panel (a) is the N12W08 flare listed. Please correct and mark the location of the flare associated with this CME. Line 100 – Incorrect, the location of the flare does not agree with the figure 1.

**Our reply:** It was corrected, and the flare locations were marked with black arrows. In Fig. 2, the location of the flare is not shown. Our main aim was to indicate the M-type flare in panel b of Fig. 2 and also the second picture of panel (a) of Fig. 1 (with a black arrow).

A major concern in this article is the use of language that does not indicate a scientific statement. 'kind of a limb CME' – is not a scientific way of describing the observations. Based on the width/ position angle of the CME please say if it is a limb CME or not.Line 107 – Again, the use of 'kind of fast' should be removed. This is a moderately fast CME. Line 110 – saying that a CME is 'somehow faster than the first CME' should be removed. The usage of words like, 'somehow', 'kind of' is not scientific and needs to be removed from the article. Please write statements that are more concrete. General statement like – 'jumps should be considerable enough' are not useful for a reader. – Please state what is considered 'considerable enough'.

**Our reply:** Thanks for your attention. All of these statements were edited in the new version of the manuscript. Besides, all of the manuscript was also edited.

Line 94 – This is incorrect – the first image in each panel shows the height-time plot for CME and not the flare observation or type. The flare observations are from the GOES satellite not LASCO. Please rewrite and correct the description of the figure and instruments used.

**Our reply:** There are GOES X-ray intensity plots below the CME height-time plot from the CDAW catalogue.

What is the source of the daily graphs of the solar wind plasma and magnetic field characteristics? Is OMNI data used, or Wind/ ACE? Please cite the correct data source. In Figure 5 – All panels include symbols extracted from OMNI data connected by a line. However, panel b – B[nT] plot shows a continuous line (not symbols and connecting line). Do the authors change the way the data is used to make this portion of the plot. Please be consistent. What are the dotted lines representing in these plots?

**Our reply:** We used the Heliospheric Shock Database generated and maintained at the University of Helsinki, which has this link I cited:  http://ipshocks.fi/database.

There are daily graphs of the solar wind and the following properties with different spacecraft, including ACE, Wind, OMNI, etc. The figures that I used were downloaded from this page, and one of them (panel a of Fig. 5) is from OMNI and panel (b) of Fig. 5 is from Wind.  of these figures are on the mentioned page, and the spacecraft names are mentioned at the top of the figures. We did not plot these figures and never changed any data. You can find these figures from the mentioned link (  http://ipshocks.fi/database) that depicted these figures with the definite spacecraft at the definite time. I used PNG plots for these figures, but you can find the original figures there. The dotted lines indicate the shocks as the page described. Again, I mentioned that these plots are not my plots; they were downloaded from the link mentioned.

Line 165 – Figure 6 displace the magnetic map --- No, the figure 6 does not display the magnetic map that is used as input. It shows the EUHFORIA boundary conditions NOT the GONG input map.

At the 0.1 au limit, 'scientists' determine the speed of the solar wind – the WSA model provides these speeds at 1 au based on an empirical formula. Please describe it correctly. What is the outer boundary? Isn't it 0.1 au? This statement is misleading.

**Our reply:** The coronal model provides the required MHD input quantities at 21.5 Rs for the heliospheric solar wind module. The coronal module in EUHFORIA is data-driven and combines a PFSS magnetic field extrapolation from GONG or ADAPT magnetograms (1-2.5 Rs) with the semi-empirical Wand-Sheely-Arge (WSA) model and the Schatteen current sheet (SCS) model to extend the velocity and magnetic field from the source surface at 2.5Rs to 21.5 Rs. This is done with other semi-empirical formulas so that the density and temperature are also derived at 21.5 Rs. EUHFORIA consists of two parts: a coronal domain and a heliospheric domain. The coronal part is a 3D semi-empirical model based on the WSA model, which provides the solar wind plasma conditions at the inner boundary of EUHFORIA at 0.1 AU. The photospheric magnetic field drives it via synoptic magnetogram maps. Figure 6 shows the heliosphere EUHFORIA  boundary condition obtained from the GONG map input and the coronal model in EUHFORIA.

Conclusion – Point 1 – self consistent structure – no observation or model comparison is shown to substantiate this claim. Point 4 – what observation data was used to calculate the arrival time , is it the linear CME velocity fit to LASCO observations ? No discussion of the interaction is included for the two CMEs. At what distance did they interact? Infact, the authors state that CME 1 and 2 arrive at different time at 1 au. If cme2 was weaker, it never overtook cme 1 , so was there any interaction? Line 218 – Figure 7 shows the result at a time instant of the solar wind AND the CMEs . It is not 'specifically focusing on the simulation of the solar wind'. This figure ONLY shows the CME evolution in the heliospheric part of EIHFORIA, without describing how the parameters of the CME model were set, tested or validated. In the text it was mentioned that CME2 is faster and expected to produce strong geomagnetic storm. And in lines 245-250 , the authors say that CME2 has diminished and less powerful. What is the reason for this?

**Our reply: Point 1:** I use a self-consistent structure due to the agreement of the overall CME shape of the simulation and observation and especially the arrival time of the CME to the Earth obtained by the simulation and observation velocity value. But if you disagree, I can omit this term.

Point 4: Yes, it was calculated using the linear fit speed based on the SOHO LASCO CME CATALOG.

CME interaction occurred for CME1 because of the other CMEs around CME1. In the CME2 case 2, no other CMEs interacted with CME2. Indeed, we do not see an interaction between CME1 and CME2. CME2 starts only when CME1 arrives at Earth.

Yes, you are right. The CME parameters from the DONKI Catalog are used. I can add the selected CME properties to the manuscript in a table. For the CME speed, we use the SOHO LASCO CME CATALOG, which indicates that the speed of CME1 is 1250 km/s and the speed of CME2 is 1450 km/s. Comparing panels b and d of Figure 8, it was concluded that CME2 has diminished ( it has a negligible size).

Line 229 – 'due to the development of magnetic field' – This description of the force dynamics is confusing. Explain the 'shrinking and expanding ' – magnetic field exists and Lorentz force is dominant in the initial phase of the CME causing acceleration before the drag takes over. Drag can accelerate or decelerate depending on the relative speeds of the CME and the solar wind.

**Our reply:** I must thank you for your help. I applied the suggestions.

Line 330 –Once the CME 1 travels out, it clears out a lot of the solar wind plasma, leaving a less dense environment into which the CME 2 will travel. So, how is the solar wind identical? Is it in the code?

**Our reply:** The background solar wind properties are the same for both CME simulations. We used the DONKI CME catalog for the CME input parameters. The properties of the background solar wind and the CMEs are illustrated in Figure 6.. Thank you for noting the important point about the differences in the pattern of the plasma density around CME2 and CME1. This would be due to the interaction of the background solar wind with the previous CMEs.

Technical :

Full form of Multi-VP

**Our reply:** MULTi-VP is not the abbreviation.

Please use the same nomenclature, you capitalize 'Sun' at some places, and not others.

**Our reply:** They were changed to Sun. I am sorry for that mistake.

Remove combining - combination– 'In this line '

**Our reply:** Combining was changed to combination.

Magnetism – magnetic field

**Our reply**: It was edited.

Please add full forms of acronyms at the first instance they appear. – SWPC,

**Our reply:** Yes, you are right. It was an abbreviation for Space Weather Prediction Center, so it was added.

Line 99 – repeated full form of STEREO

**Our reply:** It was deleted.

Line 105 – In according -> According to ..

**Our reply:** It was changed.

Line 109 – near one day -> within one day

**Our reply:** I must thank you for your attention, it was changed.

Line 117 – Why is Coronal Mass Ejections spelled ot here, when it has already been abbreviated in the beginning of the article?

**Our reply:** Yes, you are right. I apologize for that mistake. It was edited.

Time-dependent – time-dependence

**Our reply:** I am sorry. It was edited.

Line 239 – ' We determine that CME 1 will reach the Earth in approximately 46 hours ' - is this estimation based on the arrival of CME 1 in the EUHFORIA model? So, your simulated CME reaches 1 au in 46 hours?

**Our reply:** Yes, we obtained this time by following the CME1 and their time by our simulation and found approximately how long it takes to pass 1 AU (as mentioned in Figures 7 and 8, circles depict heliocentric radii at r=0.5, 1, 1.5, and 2AU).

It would be useful to indicate the CME 1 and 2 in the Figures 7 and 8 , so that readers can identify their evolution.

**Our reply:** Yes, you are right. Thank you for your attention. It was applied.

Line 150 - Add a reference here.

**Our reply:** We agree. It was added.

Add a reference for Line 59.

**Our reply:** We agree. It was added.

In Figure 7, it would be useful if the CME 1 and CME 2 are highlighted.

**Our reply:** Yes, thanks. It was applied.

////////////////////////////////////////////////////////////

Line 213 – 'Only methods related to space weather …'.    What methods?

**Our reply:** I am sorry for that mistake. It was completed.

////////////////////////////////////////////////////////////

When an equation is listed, please describe all the variables associated with the equation.

**Our reply:** We agree.  It was applied.